# Massive MIMO Techniques for 5G and Beyond—Opportunities and Challenges

**David Borges** [1,2,*], **Paulo Montezuma** [1,2], **Rui Dinis** [1,2,*] and **Marko Beko** [2,3,4]

1   Faculdade de Ciências e Tecnologia, Universidade Nova de Lisboa, 2829-516 Caparica, Portugal; pmc@fct.unl.pt
2   Instituto de Telecomunicações, Av. Rovisco Pais 1, 1049-001 Lisboa, Portugal; beko.marko@ulusofona.pt or marko.beko@tecnico.ulisboa.pt
3   COPELABS, Universidade Lusófona de Humanidades e Tecnologias, Campo Grande 376, 1749-024 Lisboa, Portugal
4   Instituto Superior Técnico, Universidade de Lisboa, 1049-001 Lisbon, Portugal
*   Correspondence: d.borges@campus.fct.unl.pt (D.B.); rdinis@fct.unl.pt (R.D.)

**Abstract:** Telecommunications have grown to be a pillar to a functional society and the urge for reliable and high throughput systems has become the main objective of researchers and engineers. State-of-the-art work considers massive Multiple-Input Multiple-Output (massive MIMO) as the key technology for 5G and beyond. Large spatial multiplexing and diversity gains are some of the major benefits together with an improved energy efficiency. Current works mostly assume the application of well-established techniques in a massive MIMO scenario, although there are still open challenges regarding hardware and computational complexities and energy efficiency. Fully digital, analog, and hybrid structures are analyzed and a multi-layer massive MIMO transmission technique is detailed. The purpose of this article is to describe the most acknowledged transmission techniques for massive MIMO systems and to analyze some of the most promising ones and identify existing problems and limitations.

**Keywords:** massive MIMO; transmission; detection; spatial multiplexing; multi-layer MIMO; power and spectral efficiency; 5G



## 1. Introduction

Wireless communications and especially mobile communication systems are growing at an incredible pace. The radio communication systems must meet a growing number of users and increasing demand for new applications, new traffic types, and data services. As an example, machine-to-machine communications will support concepts such as the smart grid, smart homes and cities, and e-health, and these applications have very diverse communications requirements, needed for a unified wireless technology to work seamlessly.

Considering the wireless data traffic, one of the key parameters to consider is wireless throughput (bits/s), which is defined as:

$$\text{Throughput (bits/s)} = \text{Bandwidth (Hz)} \times \text{Spectral efficiency (bits/s/Hz)}. \tag{1}$$

In order to improve the throughput, the bandwidth and the spectral efficiency aspects should be exploited. Increasing the bandwidth has its drawbacks, in terms of reducing the Signal-to-Noise Ratio (SNR) by Hertz for the same transmitted power, which justifies the focus of the current works on techniques that improve the spectral efficiency. A well-known way to increase the spectral efficiency is using multiple antennas at the transceivers [1]. In this context, Multiple-Input Multiple-Output (MIMO) communication systems based on the use of an antenna array at the transmitter and receiver, as illustrated in Figure 1, can offer high-speed transmission with a minimum quality of service guarantee. Energy efficiency is another aspect whose importance has been increasing along with the demand for wireless

systems. As these systems expand in different domains (power, antennas, terminals, base stations), power consumption would have grown in an unacceptable manner by using the classical techniques. New architectures and approaches are being developed with the energy efficiency goal [2–6].

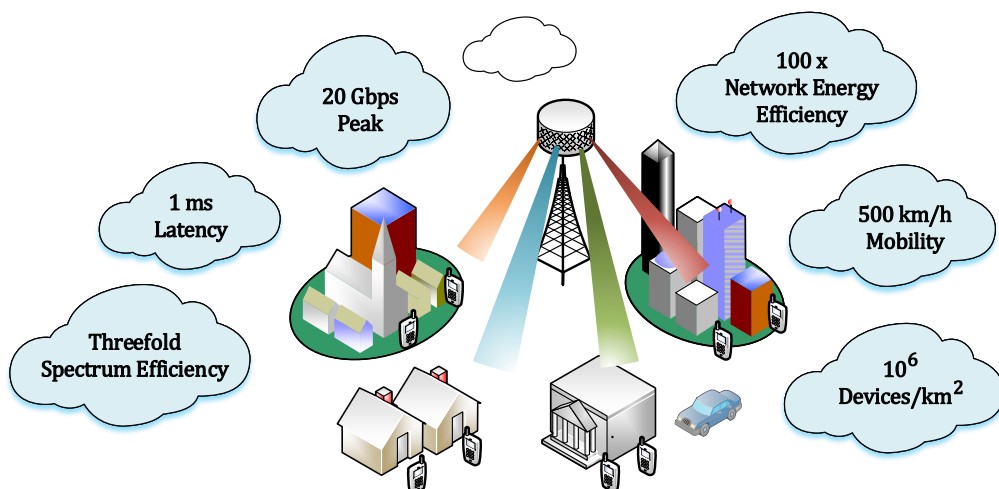

**Figure 1.** 5G major goals according to IMT-2020 and massive MIMO deployment illustration.

The spark in the research and development of MIMO systems was inspired by the work of Foschini [7] and Telatar [8], with results showing a steep linear increase in channel capacity with the increasing number of antennas being presented. MIMO systems allow us to operate two distinct dimensions of a radio link, the first being the diversity and the second being the capacity. Both dimensions have been the subject of research and developments in the recent years: the diversity improves the reliability of the communicating channel by taking advantage of multiple antenna links and the capacity can be increased by using multiple antennas and multiplexing techniques, the information transmitted being maximized through that channel. In addition, instead of increasing the system performance, the total transmitted power can be made almost inversely proportional to the number of transmitting antennas [9].

New concepts like Machine to Machine communications (M2M) and Internet of Things (IoT) in a fully connected wireless network, together with the dissemination of services like video on demand or 3D video and augmented reality consumed by mobile users, are contributing to the exponential increase of data traffic in wireless cellular networks. In fact, in the next few years, an increase of 1000 times in data traffic in wireless mobile networks is expected to respond to this global demand. To support such demand, the cellular network must dramatically increase its capacity, and using more bandwidth or increasing the network cell density will not give a viable answer to it. Another problem lies in the energy needed to support the high bit rates and the Quality of Service (QoS) of these broadband communication links. Massive MIMO technology provides a solution to increase the wireless throughput without using more bandwidth or increasing the number of cells. Massive MIMO is an extension of the MIMO technology, based on the use of hundreds or even thousands of active communicating antennas to improve spectral efficiency and throughput. It should be noted that nowadays radio networks represent more than 60% of the electricity consumption of the mobile networks and, with 5G, the trend is a significant increase in electricity consumption. Thus, even the massive MIMO solution comes with an inevitable energy problem, it being critical to assure energy efficiency while service requirements are met.

This paper presents an overview of some of key enabling technologies required to answer the huge data demand anticipated for 5G and 6G networks, highlighting the transmission techniques required for massive MIMO implementation. We start by summarizing MIMO techniques as well as massive MIMO and its premises. An analysis follows of

Single Carrier with Frequency Domain Equalization (SC-FDE) and Orthogonal Frequency Division Multiplexing (OFDM) transmission combined with MIMO and massive MIMO schemes. We discuss the concepts and the fundamental challenges underlying precoding, hardware, and receiver complexities and energy efficiency in a massive MIMO system and discuss some state-of-the-art solution techniques, limitations, and existing open problems to face in the near future. To conclude, future working directions are pointed out. The paper presents a pragmatic review of classic and novel transmission techniques for massive MIMO with a detailed description of energy efficient architectures as well as the characterization of the physical layer security introduced by layered multi-antenna architectures.

*Notation*

The following notation is used throughout this paper: bold face letters ($\mathbf{A}, \mathbf{a}$) are used to denote vectors and matrices, plain letters ($a, A$) are used to denote scalar values, uppercase letters ($\mathbf{A}, A$) refer to variables in the frequency domain, and lower-case letters ($\mathbf{a}, a$) specify variables in the time domain. Using this notation, $|a|$ is the magnitude of a scalar, $||\mathbf{a}||$ is the vector norm, $rank(\mathbf{a})$ denotes the matrix rank, $\mathbf{a}^H$ the Hermitian (conjugate transpose), $\mathbf{a}^T$ the transpose, $\mathbf{a}^{-1}$ the inverse matrix, $\mathbf{a}_k$ is the $k$th entry of $\mathbf{a}$ and the element of row $j$, and column $k$ of matrix $\mathbf{a}$ is denoted by $\mathbf{a_{j,k}}$. The identity matrix is $\mathbf{I}$. The symbol $\mathbb{E}$ is used to denote expectation.

## 2. MIMO Techniques

In wireless communication, the transmitted signals are mostly attenuated by fading, related with the multipath propagation and shadowing due to large obstacles between the transmitter and the receiver, presenting a fundamental challenge for reliable communication. Transmission with multiple antennas is a well-known diversity technique to improve the resilience of the communication when the same signal is transmitted or received by multiple antennas. Another possible approach by employing multiple antennas is to transmit multiple data streams and hence obtain a multiplexing gain that improves the communication throughput in a significant way. In the past decades, MIMO systems have gained substantial attention and are now being incorporated into several new generation wireless standards.

By employing numerous antennas in the communication system, the transmission channel becomes rich in links between communicating antennas. Considering that the antennas in the array are separated enough to have independent channel realizations, each signal observed at the receiver will have traveled through an independent Rayleigh fading channel [10] from each transmitting antenna and have an independent channel gain. This antenna separation was defined to be $\lambda/2$ as described in [11]. In this scenario, illustrated in Figure 2, if a path between communicating antennas is faded or even blocked, other path(s) can still be available for transmitting a signal from the transmitter to receiver. This technique, regarded as spatial diversity, presents numerous freedom degrees that can be exploited to increase the system performance [12].

Since the transmitted signal reaches the receiver through many different paths, the superimposed received signals can either reinforce or cancel each other. When the number of added signals is large, the central limit theorem can be taken into account and a Gaussian distribution can be considered as an approximation [13]. This phenomenon is a microscopic effect caused by small variations in the propagation environment (e.g., movement of the receiver, transmitter, or other objects) regarded as small-scale fading. On the other side, variance is interpreted as the macroscopic large-scale fading, which includes antenna gains, shadowing, pathloss, and penetration losses in Non-Line-of-Sight (NLoS) propagation [10].

The Alamouti's MIMO scheme is a transmit diversity scheme for two transmit antennas that does not require transmit channel knowledge, and it can be considered as the simplest diversity Space Time Block Code (STBC) scheme. Despite its simplicity, by only using two-transmit antennas and a simple signal processing at the receiver, it provides a clear improvement on the received signal. The diversity order obtained with this technique

is the same achieved by Maximum Ratio Combiner (MRC) with two receiving antennas. It is also possible to extend to the case of two-transmit antennas and N receiver antennas, which would provide a diversity order of 2N [14].

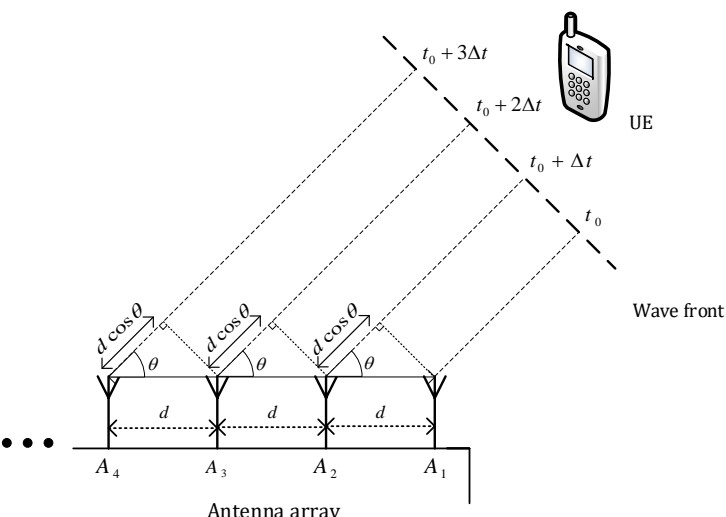

**Figure 2.** Multiple antenna transmission scenario.

The benefits of having multiple antennas can be further improved when the transceiver knows the channel response of the communication link. This information enables the receiver to coherently combine the received signals from all antennas, a technique regarded as receive combining. If this information is present on the transmitter side, the signal can be pre-processed to lower the equalization complexity on the receiver side [15]. This later technique is called precoding. Estimation of the channel response is thus a key aspect in multi-antenna systems. In addition, the use of multiple communicating antennas on the transmitter's side can be employed to achieve beamforming where a constructive interference of two or more electromagnetic waves are added coherently and produce a higher amplitude signal at a desired point in space [16].

By using multiple transmit antennas to send copies of the same signal, it is expected that we will have many observations and can better decide about the estimated transmitted signal. This is the principle behind the transmit diversity technique, which can be extrapolated to receive diversity, where multiple antennas are employed on the receiver side in order to acquire multiple measurements of the same signal [17]. All of these techniques are able to provide significant improvements of the system performance, at the cost of additional signal processing, size of the transmitter/receiver, energy consumption, and the actual price of the antennas [1]. The balance of system performance against the available resources needs to be evaluated when choosing the appropriate scheme.

*2.1. Channel Capacity*

Capacity is one of the main measurements to characterize the performance of wireless systems, and it also practically serves as a guide to properly design the transmitted signals as well as the processing of the received signals [18]. Different implementation regarding the use of multiple antennas was considered in order to study the most efficient capacity increase. Antennas can be added to the transmitter, to the receiver, or to both sides of the transmission system. As is known from information theory, the notion of channel capacity was introduced by Claude Shannon by an expression often known as Shannon's capacity:

$$C = B \cdot \log_2 \left( 1 + \frac{p|h|^2}{B \cdot n_0} \right), \tag{2}$$

where $c$ is the channel capacity in bits per second, $b$ is the bandwidth of the channel in hertz, $p$ is the symbol power, $h$ is the channel gain, and $n_0$ is the noise variance [19]. Considering

the theoretical upper limit, where $\mathbb{E}[|h|^2] = 1$, and removing the fixed channel bandwidth, we can then simplify the expression as

$$C = \log_2(1 + \text{SNR}), \tag{3}$$

where the capacity $C$ is now measured in bits/hertz and $\text{SNR} = \frac{p}{n_0}$ is the signal-to-noise ratio at the receiver.

### 2.1.1. SISO

The case where we have a single communicating channel is regarded as Single-Input Single-Output (SISO), and this scenario is depicted in Figure 3, where the information signal $x$ is convoluted with the channel impulse response $h$ in the time domain and the Gaussian noise $n$ is added afterwards. In the frequency domain, the received signal $Y$ is given by

$$Y = HX + N, \tag{4}$$

where $Y$ denotes the Fourier transform of the received signal $y$, $H$ is the channel's frequency response, $X$ the Fourier transform of the transmitted signal $x$, and $N$ denotes the noise. With only one transmitting and one receiving antenna, there is only one communication channel and all the variables are scalars, so the system capacity is given by Equation (3).

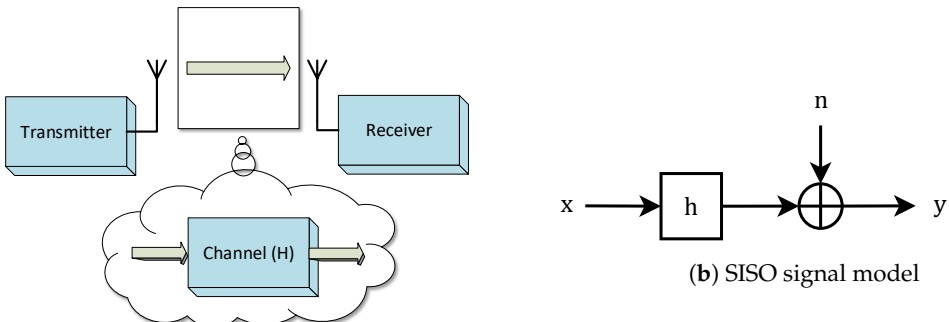

(**a**) SISO channel model

(**b**) SISO signal model

**Figure 3.** SISO model architecture.

### 2.1.2. SIMO

The addition of multiple antennas to one side of the communication system can be regarded as Single-Input Multiple-Output (SIMO) when antennas are added on the receiver side, or Multiple-Input Single-Output (MISO) when antennas are employed on the transmitter side. Adding $R$ antennas to the receiving side will result in $R$ channel links, where $h_i$ is the complex gain between each transmitting and receiving antenna.

In the SIMO case, represented in Figure 4, the received signal $y_r$ can be defined as the convolution of $x$ with the respective channel $h_r$ and the Gaussian noise $n_r$ summation. Having $R$ receiving antennas, each received signal can be described in the frequency domain as

$$Y_r = H_r X + N_r, \tag{5}$$

$N_r$ and $N_r$ being the channel frequency response and noise between the transmitting antenna and the $r$th receiving antenna, respectively. It is convenient to compact the multiple signals and represent them by vectors as

$$\mathbf{Y} = \begin{bmatrix} Y_1 \\ \vdots \\ Y_r \end{bmatrix}; \quad \mathbf{H} = \begin{bmatrix} H_1 \\ \vdots \\ H_r \end{bmatrix}; \quad \mathbf{N} = \begin{bmatrix} N_1 \\ \vdots \\ N_r \end{bmatrix}. \tag{6}$$

This representation allows us to write the received signal vector **Y** in the frequency domain as

$$\mathbf{Y} = \mathbf{H}X + \mathbf{N}, \tag{7}$$

where **H** is the $R$-dimensional vector representing the channel frequency response, $X$ the information symbol, and **N** the $R$-dimensional noise vector. From Equation (7), we can conclude that the received signal vector is the sum of two vectors: the scaled channel vector plus the noise vector. Considering this geometric point of view, we can conjecture that the received vector **Y** should be evaluated in the direction of the vector **h**. This geometric approach is illustrated in Figure 5.

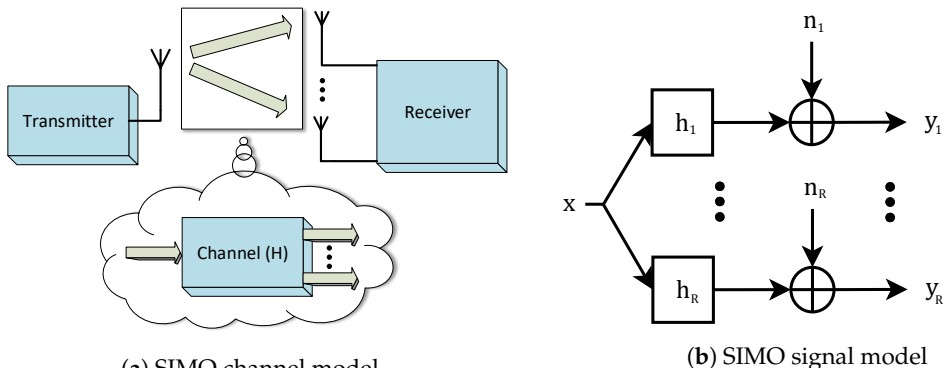

(**a**) SIMO channel model  (**b**) SIMO signal model

**Figure 4.** SIMO model architecture.

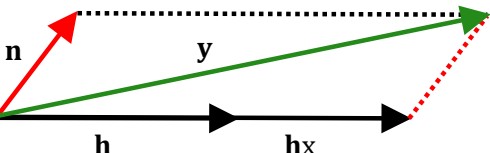

**Figure 5.** Geometric representation of SIMO signals.

We should note that the direction of these vectors has no physical relation with the direction in space (at least a simple one), as we speak about an $R$-dimensional vector.

On the receiver side, we intend to recover the information symbol $x$ based in the signal vector **y**. Since we know that the information must be in the direction of the channel vector, everything pointing in another direction than **h** can only be considered as noise. Still, part of this noise can point in the channel direction. For this purpose, we should project the vector **y** into the direction of **h**, represented mathematically as the inner product of vectors. This result will then be normalized with respect to the norm of **h**. In wireless communications, this technique is regarded as receive combining. For this example, we will describe the combining technique regarded as MRC to maximize the SNR at the receiver. This and other combining methods will be further analyzed in Section 5.

In terms of the capacity, by applying MRT precoding [20], we can formulate the channel capacity of the SIMO system as

$$C = \log_2\left(1 + \frac{p\|\mathbf{h}_i\|^2}{n_0}\right), \tag{8}$$

where the squared norm of the channel vector $\|\mathbf{h}_i\|^2$ is defined as the sum of the squared absolute value of each channel gain, so that we can re-write the previous expression as

$$C = \log_2\left(1 + \frac{p}{n_0}\sum_{r=1}^{R}|h_r|^2\right). \tag{9}$$

We should note that the $R$ noise terms are added not constructively but randomly, so we have the same noise variance $N_0$. Considering the upper limit where all the channels have the same unit gain, the capacity formula can be simplified as

$$C = \log_2(1 + R \cdot \text{SNR}). \tag{10}$$

We can conclude that a $R$ times stronger signal is received by employing $R$ receiving antennas and a combining algorithm on the receiver side. This gain is designated as spatial diversity gain as it arises from the spatially separated receiving antennas.

### 2.1.3. MISO

The next scenario is characterized by multiple antennas on the transmitter's side and a receiver with only a single antenna, as shown in Figure 6.

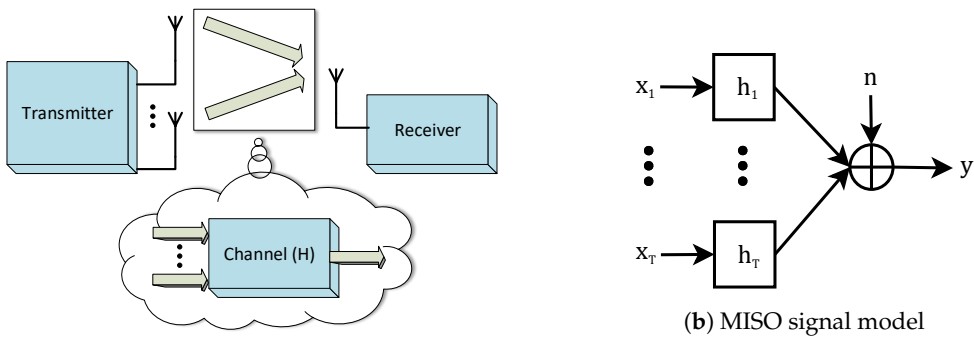

(**a**) MISO channel model

(**b**) MISO signal model

**Figure 6.** MISO model architecture.

In this model, the $T$ multiple antennas will be fed with a signal vector $\mathbf{X} = [X_1, X_2, \ldots X_n]$ where each input $X_t$ will be multiplied with the respective channel frequency response $H_t$. All of these signals will be then added together at the receiver along with one noise scalar term $N$. We can represent the received signal in the frequency domain as

$$Y = \sum_{t=1}^{T} H_t X_t + N. \tag{11}$$

Again, it is convenient to represent this model using vectors, so we have

$$\mathbf{X} = \begin{bmatrix} X_1 \\ \vdots \\ X_t \end{bmatrix}; \quad \mathbf{H} = \begin{bmatrix} H_1 \\ \vdots \\ H_t \end{bmatrix}, \tag{12}$$

where $\mathbf{X}$ is the $T$-dimensional information vector and $\mathbf{H}$ is the $T$-dimensional channel vector containing each channel frequency response $\mathbf{H_t}$. Thus, the received signal can be re-written as

$$Y = \mathbf{H}^T \mathbf{X} + N. \tag{13}$$

We can do the same geometric analysis as in the SIMO case and notice that, by doing the inner product $\mathbf{h}^T \times \mathbf{x}$, the information vector is being projected to the channel vector direction. This means that there is no point in sending information in any 'direction' other than the channel vector. Having this information in mind, we can design the information vector to match the channel vector direction. This signal processing technique is regarded as precoding. The precoding vector, defined as $\mathbf{w}$, is multiplied by the information symbol $\tilde{x}$, so that we have

$$\mathbf{x} = \mathbf{w}\tilde{x} \tag{14}$$

**x** being the effective transmitted vector and **w** the *T*-dimensional precoding vector for the MISO case. This architecture is shown in Figure 7.

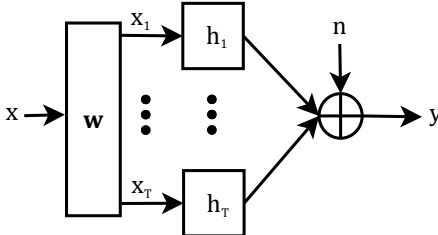

**Figure 7.** Precoded MISO signal model.

By employing a similar technique as the previous scenario, regarded here as the Maximum Ratio Transmitter (MRT), the precoding vector **w** is defined as

$$\mathbf{w} = \frac{\mathbf{h}^*}{||\mathbf{h}||}. \tag{15}$$

Thus, the received signal can be then re-written as

$$Y = \mathbf{H}^T \mathbf{W} \tilde{X} + N = ||\mathbf{H}|| \tilde{X} + N, \tag{16}$$

and the MISO system capacity will be formulated as

$$C = \log_2 \left( 1 + \frac{p}{n_0} \sum_{i=1}^{T} |h_i|^2 \right) \tag{17}$$

which can be simplified as

$$C = \log_2 (1 + R \cdot \text{SNR}) \tag{18}$$

For both cases, the SNR increases linearly with the number of additional antennas. It can be observed that this gain is achieved in different ways: In the SIMO case, multiple signal observations are constructively combined at the receiver, so this gain is achieved through receiver spatial diversity; in the case of MISO, the signal is previously adapted to the channel to assure a strong signal directivity. This technique is labeled as precoding, more precisely as beamforming.

### 2.1.4. MIMO

For the case of a MIMO channel of Figure 8, we consider *T* transmitting antennas and *R* receiving antennas.

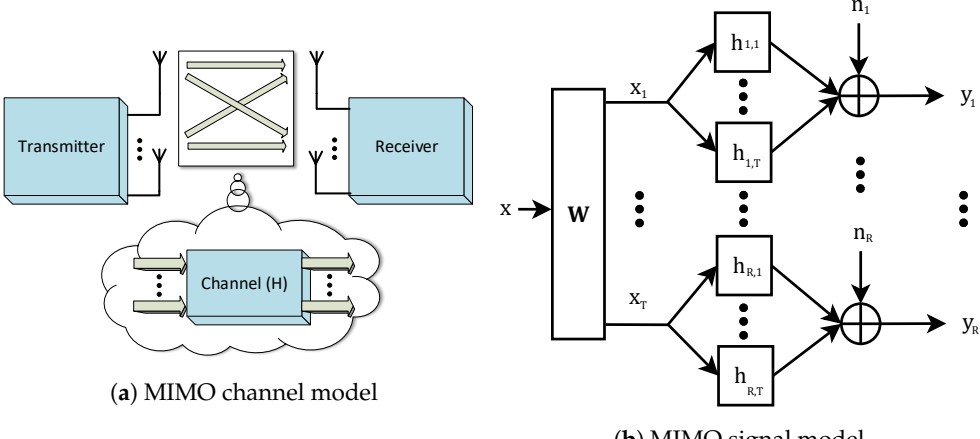

(**a**) MIMO channel model

(**b**) MIMO signal model

**Figure 8.** MIMO model architecture.

As shown in Figure 8b, the complex gain $h_{r,t}$ corresponding to the channel response from the $t$th transmitting antenna to the $r$th receiving antennas is denoted as $h_{r,t}$. Each signal $x_t$ will be transmitted from one antenna to $R$ receiving antennas and so the frequency domain received signal at each receiving antenna $Y_r$ can be formulated as

$$Y_r = \sum_{t=1}^{T} H_{r,t} X_t + N_r. \tag{19}$$

As we will have $R \times T$ channel responses and multiple signals, it is convenient to represent this model using vectors and matrices. The model can be then represented with

$$\begin{bmatrix} Y_1 \\ \vdots \\ Y_R \end{bmatrix} = \begin{bmatrix} H_{1,1} & \cdots & H_{1,T} \\ \vdots & \ddots & \vdots \\ H_{R,1} & \cdots & H_{R,T} \end{bmatrix} \begin{bmatrix} X_1 \\ \vdots \\ Y_T \end{bmatrix} + \begin{bmatrix} N_1 \\ \vdots \\ N_R \end{bmatrix} \tag{20}$$

or simply by

$$\mathbf{Y} = \mathbf{H}\mathbf{X} + \mathbf{N}, \tag{21}$$

where $\mathbf{Y}$ is the $R$-dimensional received signal vector, $\mathbf{H}$ the $R \times T$ channel matrix of frequency responses, $\mathbf{X}$ the $T$-dimensional vector defined as the FFT of the precoded signal $\mathbf{x}$, and $\mathbf{N}$ the $R$-dimensional noise vector. Considering the previous scenarios, it is straightforward that some signal processing will be needed on the transmitter and receiver sides in order to deal with the multiple signals. It was showed in [21] how to compute the Singular Value Decomposition (SVD) of the channel matrix as $\mathbf{h} = \mathbf{u}\mathbf{s}\mathbf{v}^H$ the matrices $\mathbf{u}$ and $\mathbf{v}$ represent the left and right singular vectors of $\mathbf{h}$, respectively. Matrix $\mathbf{u}$ can be employed as the combining matrix and $\mathbf{v}$ as the precoding matrix, so we can write $\mathbf{x} = \mathbf{v}\tilde{\mathbf{x}}$ and $\tilde{\mathbf{x}} = \mathbf{u}^H \mathbf{y}$. By employing this signal processing method, the MIMO channel becomes diagonal, and it is converted into a number of parallel SISO channels with different effective SNRs, related to the square of the singular value of each channel. The number of parallel channels is defined by $S = \text{rank}(\mathbf{h})$, which is smaller than $\min(R, T)$. Since there is no more cross-talk between these parallel links, each link contributes individually to the overall capacity as an isolated link. We already defined the capacity of the SISO channel, so we can formulate the capacity of each sub-channel as

$$C = \log_2 \left( 1 + \frac{q_k s_k^2}{N_0} \right) \tag{22}$$

where $q_k$ is the transmit power, $s_k$ the respective singular value, and $N_0$ the noise power spectral density. As there are $S$ parallel channels, the total capacity will be the sum of each capacity. In order to maximize it, the power allocated to each sub-channel needs to be optimized, having in mind the following capacity equation:

$$C = \max_{q_1 \geq 0, \dots, q_S \geq 0} \log_2 \left( 1 + \frac{q_k s_k^2}{N_0} \right)$$
$$\text{s.t.} \sum_{k=1}^{S} q_k = q \tag{23}$$

It was already shown in [22] that the optimal power allocation is given by the water-filling method, which will assign higher powers to channels with better conditions. It can be defined as

$$q_k^{opt} = \max \left( \mu - \frac{N_0}{s_k^2}, 0 \right). \tag{24}$$

The MIMO capacity can then be re-written as

$$C = \sum_{k=1}^{S} \log_2 \left( 1 + \frac{q_k^{opt} s_k^2}{N_0} \right) \tag{25}$$

which in the case of equal singular values can be simplified as

$$C = S \cdot \log_2(1 + SNR). \tag{26}$$

The MIMO gain is achieved through a multiplexing gain, which means that, if the propagation channel has enough freedom degrees, the capacity scales linearly with $\min(R, T)$ and logarithmically with SNR, which can lead to enormous capacity gains when the number of antennas is large. However, as the propagation channel will act as the limiting factor, understanding the channel is essential for bringing the theoretic gains into practical systems [23].

## 2.2. Signal Modulation

One way to increase the capacity of a given system by employing multiple antennas at both the transmitter and the receiver was described here. To exploit this potential, layered space–time architectures have been proposed for flat fading MIMO channels [21], and these techniques were later extended to frequency-selective channels using time-domain Decision Feedback Equalizers (DFE) in MIMO systems [24].

Nonetheless, for the high information rates of broadband wireless frameworks, we can have extreme time-scattering impacts related with the multipath propagation. For this situation, ordinary time-domain equalization techniques are not adequate, since the number of operations per symbol is proportional to the Inter-Symbol Interference (ISI) length. This effect can be stronger when ordinary time-domain equalization techniques are utilized in high information rate MIMO frameworks [25]. In addition, the loss of orthogonality between users in severe time-dispersive channels can lead to a significant performance degradation, creating the need for high-complexity receiver structures.

Block transmission techniques combined with FDE techniques and appropriate cyclic extensions have been shown to be adapted for high data rate transmission over severe time-dispersive channels, since the number of operations per symbol only grows logarithmically with the block duration (and, therefore, the ISI span), due to the Fast Fourier Transform (FFT) implementation. One of the most popular modulations based in the block transmission technique is OFDM. The transmission of OFDM signals over a wideband time-dispersive channel can be considered as a parallel transmission over a series of narrowband non-dispersive channels (subcarriers), so the extension of MIMO/BLAST techniques to OFDM schemes is simple, perhaps with additional pre-processing and/or employing adaptive loading schemes. However, OFDM signals carry high envelope fluctuations and a high Peak-to-Average Power Ratio (PAPR) leading to amplification drawbacks [26]. Because of this, numerous techniques for reducing the envelope fluctuations of OFDM signals have been proposed [27–29]. However, these techniques require an increased number of signal-processing tasks, especially on the transmitter side, and possibly some signal distortion when a nonlinear (NL) signal processing is employed, such as the amplitude clipping technique [30–32]. Despite the additional complexity, the PAPR reduction techniques do not achieve a null PAPR or a value near zero, essential to maximizing the efficiency of RF signal power amplification. Thus, even for the most sophisticated PAPR reduction techniques, the transmitted signals still have PAPR higher than those for SC signals based on similar constellations, which makes an efficient amplification difficult.

Single Carrier (SC) modulations, combining block transmission techniques and FDE, are an alternative approach for broadband wireless systems. Like in OFDM modulations, a Cyclic Prefix (CP) is appended to data blocks, long enough to cope with the channel length. The received signal is converted to the frequency domain, equalized in the frequency domain, and then transformed back to the time domain. A simple time-frequency domain

comparison between SC and Multi Carrier (MC) modulations is presented in Figure 9. The overall implementation complexity, as well as the achievable performance, is similar for SC schemes with FDE and OFDM schemes [33]. However, the signal-processing load is more intense at the receiver for the SC case. This situation, combined with the lower envelope fluctuations of SC signals, makes them more suitable for uplink transmission (i.e., the transmission from the User Equipment (UE) to the Base Station (BS)), while the OFDM schemes prevail as a better choice for the downlink transmission (i.e., the transmission from the BS to the UE) [34].

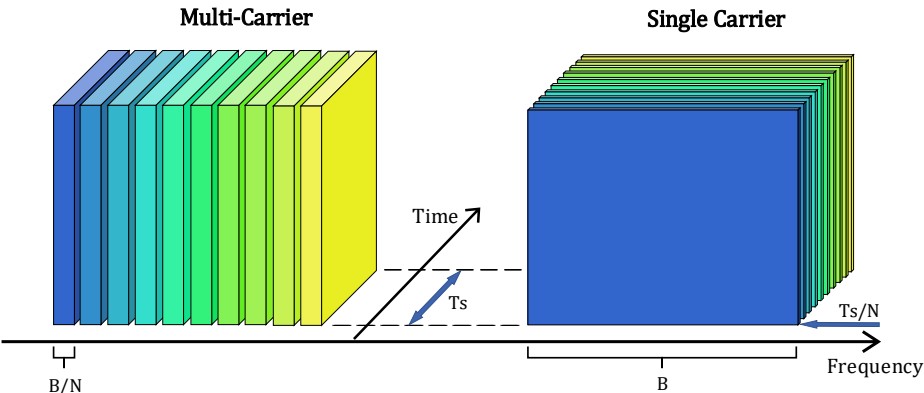

**Figure 9.** Comparison of Multi-Carrier versus Single-Carrier time-frequency arrangement.

In most of the cases, a linear FDE is used at the receiver, although it was already proved that NL equalizers can have significantly better performance than linear equalizers [35]. For this reason, it is beneficial to design NL equalizers for SC-FDE schemes [36]. Among several different NL equalizers, DFE is especially attractive due to its good performance-complexity trade-off [12]. A hybrid time–frequency SC-FDE employing a frequency-domain feedforward filter and a time-domain feedback filter were proposed [37]. Although this scheme can provide better performance than a linear FDE, it can suffer from error propagation as in the conventional time-domain version, especially if the feedback filter has a large number of taps.

A successful Iterative Block DFE (IB-DFE) approach for SC transmission was proposed in [38] and extended to diversity scenarios and spatial multiplexing schemes in [39]. The IB-DFE schemes were also tested and proved to allow excellent results in many other scenarios, spanning from network diversity schemes, Code Division Multiple Access (CDMA), reduced CP and offset modulations. Highly efficient techniques based on the IB-DFE concept were developed for joint detection and estimation and to cope with severe NL distortion effects [40]. In IB-DFE schemes, both the feed-forward and the feedback circuits are implemented in the frequency domain. The error propagation problem is therefore significantly reduced since the feedback loop considers both the hard decisions and the overall block reliability for each received block. Thus, IB-DFE techniques allow for much better performances than non-iterative methods [41,42] and can be regarded as low-complexity turbo equalization schemes, since the feedback loop uses the equalizer outputs instead of the channel decoder outputs. In the early stages of IB-DFE implementations, hard decisions were considered in the feedback loop and weighted by the block-wise reliability. To improve the system performance and allow truly turbo FDE implementations, IB-DFE schemes with soft decisions were proposed in [43]. Even though most of the IB-DFE implementations only consider the adoption of Quadrature Phase Shift Keying (QPSK) constellations, larger constellations such as Quadrature Amplitude Modulation (QAM) and M-Phase Shift Keying (MPSK) are often required when we want to increase the system's spectral efficiency [44]. Furthermore, hierarchical constellations, which may be composed of non-uniformly spaced constellation points, are particularly interesting for broadcast and multicast systems as they are able to provide high bit error protection. In this type of constellation, there are two or more classes of bits with different error protection, to which

different streams of information can be mapped. A given user can attempt to demodulate only the more protected bits or also the other bits that carry additional information, depending on the channel conditions. An application of these techniques is in the transmission of coded voice or video signals, where different error protection associated with different resolutions can exist.

In terms of the achievable data rate, which has been proven tight when the channel hardens, SC and OFDM transmission are equivalent in massive MIMO. Due to channel hardening, all tones of the OFDM transmission have equally good channels, therefore the advantage of using OFDM and employing the water filling method across frequencies results in little gain.

The PHY layer radio access network emerges as critical design for the envisioned service architecture. If synchronism is needed, the transceivers need to operate with a common clock for their processing. Furthermore, when we consider OFDM, the waveform detection process is free of crosstalk only when orthogonality is assured. Both aspects are related, such that some type of synchronization is often required to establish orthogonality. However, as soon as the orthogonality is broken (for example, due to random channel access or multi-cell operation), the signal distortion grows extremely in OFDM. [45] In fact, 5G is expected to be centered on OFDM-based schemes. Nonetheless, waveforms like Discrete Fourier Transform spread OFDM (DFT-S-OFDM), Block-Windowed Burst OFDM (BWB-OFDM), and Generalized Frequency Division Multiplexing (GFDM) include SC-FDE as special cases, with the associated advantages, especially for the uplink transmission [46,47]. Moreover, massive MIMO is not restricted to 5G, and the advantages of SC-FDE-based mMIMO schemes apply to other scenarios.

## 3. Massive MIMO

The general concept of massive MIMO is defined as a physical-layer technology which equips each BS with a huge number of active antennas that can be used to spatially multiplex many UEs so that it is possible to communicate with them on the same time-frequency resource. The spectral efficiency per cell can be improved by orders-of magnitude over classical cellular networks by coping with the signal attenuation and interference through spatial signal processing with techniques such as receive combining and transmit precoding. To resume, massive MIMO is an upgraded version of the Space-Division Multiple Access (SDMA), pushing the spatial multiplexing to an extreme level [48].

The main benefits of massive MIMO systems can be summarized as:

- Huge spectral efficiency;
- Communication reliability;
- High energy efficiency;
- Low complexity signal processing;
- Favorable propagation;
- Channel hardening.

Massive MIMO inherits all gains from conventional Multi-User MIMO (MU-MIMO), i.e., with M-antenna BS and K single-antenna users, we can achieve a diversity of order M and a multiplexing gain of min (M,K). By increasing both M and K, we can obtain a huge spectral efficiency and very high communication reliability with simple linear processing such as MRC, Zero Forcing (ZF), and Minimum Mean Square Error (MMSE).

Several asymptotic limits of random matrix theory are approximated by the law of large numbers when MIMO arrays are made large [49]. It is expected that each antenna would be contained in an inexpensive module with simple processing and a low-power amplifier. Several results can be used in the system design, as events that were random before now start to look deterministic. An example is the distribution of the channel matrix singular values that approach a deterministic function [50]. Another example is that very wide or very tall matrices tend to be very well conditioned. In addition, when dimensions are large, some matrix operations such as inversions can be done fast by using series expansion techniques. In the limit of an infinite number of antennas at the base

station, the linear processing in the form of MRC for the uplink and MRT on the downlink is optimal.

Other effect arising from the dimension increase is that the average sum of the thermal noise tends to be much lower than the overall interference levels, which means that the system is predominantly limited by interference from other transmitters. This effect is more evident for the uplink, since coherent combining performed on the receiver side reduces substantially effects that are uncorrelated between the antenna elements, in particular the thermal noise and some effects associated with imperfections.

Finally, when the size of the array grows, the spatial resolution of the array increases, so that it is possible to resolve individual scattering centers with remarkable precision. The communication performance of the array, when the number of antennas is large, depends less on the actual statistics of the propagation channel but more on the aggregated properties of the propagation such as asymptotic orthogonality between channel vectors associated with distinct terminals [23].

In the massive MIMO uplink, coherent combining can achieve a very high array gain, allowing for a considerable reduction in the transmit power of each user. In the downlink, if the BS knows the location of the user terminals, it can focus the transmitted power to the desired spatial directions. As a result, the radiated power can be greatly reduced and hence we can obtain high energy efficiency with massive antenna arrays. For example, with a fixed number of users, by doubling the number of BS antennas, the transmit power is reduced by a factor of two, while the original spectral efficiency remains unchanged, and hence the radiated energy efficiency is doubled.

For most propagation environments, the use of an extra number of BS antennas over the number of users offers favorable propagation, making the channel vectors between the users and the BS to be pair wisely (nearly) orthogonal. Under favorable propagation, the effect of inter-user interference and noise can be eliminated with simple linear signal processing in the downlink and in the uplink. As a result, simple linear processing schemes perform nearly optimally.

Under some conditions, when the number of BS antennas is large, the channel becomes close to deterministic so that the effect of small-scale fading is averaged out. The system scheduling, power allocation, and other controls can be done over the large-scale fading time scale instead of over the small-scale fading time scale.

The propagation channel **h** provides asymptotic channel hardening if

$$\frac{||\mathbf{h}_k||^2}{\mathbb{E}\{||\mathbf{h}_k||^2\}} \to 1 \tag{27}$$

as $M \to \infty$. From this definition, we can interpret that the gain $||\mathbf{h}_k||^2$ of a random fading channel $\mathbf{h}_k$ is close to its mean value when the number of antennas is very high, so that the deviation from the channel average will vanish asymptotically.

This simplifies the signal processing significantly and improves the downlink channel gain. One crucial advantage of channel hardening is that the receiver does not need instantaneous Channel State Information (CSI) to detect the transmitted signals. In this way, the system design including scheduling and resource allocations can be done regarding the large-scale fading time scale by applying the statistical knowledge of the channel gains [49]. The statistical property regarding the large-scale fading of the channel changes very slowly and hence the overhead for the channel estimation is significantly reduced. This fact is commonly used in massive MIMO via the use-and-then-forget capacity bound. Lastly, under the channel hardening effect, the main performance metrics such as the achievable rate depend only on the large-scale fading.

Massive MIMO is highly leveraged by the usage of millimeter Wave (mmWave) bands, since the decrease in wavelength enables packing large antenna arrays at both the transmitter and receiver [51]. The term mmWave assumes frequencies to be in the interval from 30 GHz to 300 GHz, so that we can measure the carrier wavelength between 10 mm and 1 mm. As the antenna arrays usually have antenna spacing of one or a half wavelength,

this allows for the addition of more active antennas per unit of area. Looking at the Friis transmission equation,

$$\frac{P_r}{P_t} = \frac{A_r A_t}{d^2 \lambda^2} \tag{28}$$

we can observe that, by transmitting the same power $P_t$, keeping the receiving and transmitting antenna effective area aperture $A_r$ and $A_t$ and reducing the carrier wavelength $\lambda$, the received power $P_r$ will decrease in a squared proportionality.

While massive MIMO is beneficial at centimeter-wave frequencies, it can be essential in the millimeter-wave bands, since the high free-space pathloss at those frequencies necessitates large array gains to obtain sufficient received SNR. In addition, these bands are less crowded than the lower frequencies used nowadays in mobile communications, making bigger slices of spectrum available. The usage of more bandwidth requires a bigger amount of power to maintain the received SNR. If the objective is not to spend more power, the communication range can be sacrificed to keep a constant received SNR value. An elegant solution for this problem which was already referred is the beamforming. With this technique, the energy is more focused in the desired receiving direction instead of an omni-directional transmission. The use of directional communications in these bands requires the employment of location, sensing, and detection techniques as analyzed in [52–54]. One issue which is conditioned by the physical laws is the worse penetration capacity of these mmWave frequencies. This issue can be tackled with a higher power transmission or following a new path and rethinking the construction materials of infrastructures and transports. However, in practice, this effect can actually be an advantage, since lower penetration means better insulation between cells and lower interference levels.

The integration of massive MIMO and mmwave technologies in legacy cellular systems presents considerable challenges, not only at the physical layer but also at the medium access level (see, e.g., [55,56] and references within). To overcome these difficulties, the access should be done in a joint way [57,58].

## 4. Implementation Difficulties

Massive MIMO technology presents some critical issues that have to be addressed. One of these issues is the performance degradation due to hardware impairments, since low-cost RF chains are employed in the system [59,60]. A large number of antennas means a high cost and power consumption for some mixed signal components, like high-resolution Analog-to-Digital Converters (ADCs), which makes it difficult to dedicate a complete RF chain for each antenna (it should be noted that Digital-to-Analog Converters (DACs) at the transmitter consume less power). Low-resolution ADCs can be implemented with simpler circuits, consume less power, and at low SNR incur only a little rate loss compared to high-resolution quantization [61,62].

Channel estimation errors and channel aging effects are other issues which can degrade the system performance, especially in fast mobility environments [63,64]. In high-speed environments, it is very likely that the channel coefficients change between the instant they are estimated and the moment they are used by the precoder or equalizer, the phenomenon regarded as channel aging. Analytical and simulation results concluded that SC-FDE is more robust to channel aging and double dispersion effects than OFDM [60]. However, when channel estimation errors exist, the sum-rate of both systems saturates as the number of BS antennas gets large.

A dynamic wireless channel needs to be estimated after every coherence interval. Massive MIMO systems were first conceived for TDD operation, where the channel is systematically learned in one direction and signal processing can be applied at both ends assuming channel reciprocity. In these systems, the channel matrix becomes of great dimensions due to the huge amount of communicating antennas [65]. Due to this fact, different techniques have been adopted in order to estimate the channel characteristics in a low-complexity manner [66,67] or by taking a physical sense approach [68].

The use of compressed sensing reduces the channel estimation requirements for MIMO and massive MIMO schemes [69,70] namely when combined with localization information [53,71].

In addition, in the physical domain, there is one undesired effect arising from the usage of multiple active antennas. Mutual coupling is the electromagnetic interaction between the antenna elements in an array. The array configuration sets the base mutual coupling, but the excitation of each antenna element and the contribution from the adjacent ones will influence the current present in the antenna element. In practice, the mutual coupling effect depends on the array configuration and the excitation of the antenna elements [72]. This effect changes the radiation patterns of the antenna array, as it alters the self and mutual impedances, resulting in a degraded radiation efficiency [73]. Although it tends to degrade the performance of the system, this effect can be exploited for array calibration. Considering the mutual coupling effect, the antenna correlation tends to reduce as its separation decreases [74]. Detailed mechanisms of mutual coupling can be developed, depending on the transmitting/receiving modes, and different approaches have been adopted to reduce the coupling effects [75,76].

The more antennas the BS is equipped with, the more degrees of freedom are offered and hence more users can simultaneously communicate at the same time–frequency resource. As a result, a huge sum throughput can be obtained. However, for large antenna arrays, the conventional signal processing techniques (e.g., maximum likelihood detection) become prohibitively complex due to the high signal dimensions. Thus, it becomes crucial to assure this huge multiplexing gain with low-complexity signal processing and low-cost hardware implementation.

Under certain circumstances, the performance of a very large array becomes limited by interference arising from re-use of pilots in neighboring cells [77]. In addition, choosing pilots in a smart way does not substantially help as long as the coherence time of the channel is finite. This effect was quantified in [78], for a Time-Division Duplex (TDD) system, under the assumption of channel reciprocity and that base stations estimate downlink channels based on uplink received pilots.

Pilot contamination is a critical problem in cellular massive MIMO, which is caused by the lack of orthogonality in pilot sequences transmitted from adjacent cells. In multi-cell systems, pilots must be reused between cells at a certain re-usage degree, which contaminates channel estimates in the home cell with pilots from adjacent cells. In this situation, the estimated channel vector in a cell is the sum of the channel vectors of users from the neighboring cells plus the intended cell. This phenomenon results in degradation of the channel estimate quality and an incoherent interference that does not disappear with the addition of more antennas. As the number of interfering cells increases, this problem grows exponentially [25].

## 5. MIMO Implementations

Fully digital beamforming, or simply digital beamforming, provides the highest flexibility in terms of the possible beamforming algorithms that can be employed. This comes from the fact that, by digitally manipulating the signal, it is possible to adjust the phase and amplitude of each signal that feeds an antenna element. This scenario, shown in Figure 10, requires each antenna connected to the baseband through a dedicated mixer, a DAC, a filter, and an amplifier, i.e., an entire RF chain. This turns the implementation of digital beamforming in a massive MIMO architecture with hundreds of antennas in an expensive and challenging task due to the high power consumption, complexity, and cost.

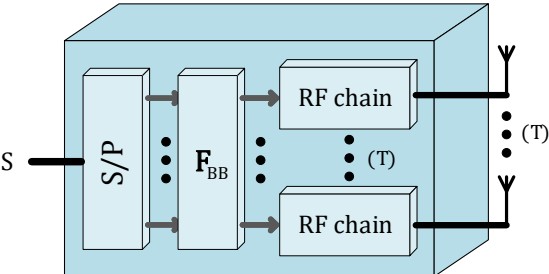

**Figure 10.** Digital beamforming structure.

Millimeter wave frequencies will probably be operated together with massive antenna arrays to overcome propagation attenuation. This makes a fully digital user separation not feasible, since the amount of energy required for all the analog-to-digital and digital-to-analog conversions would be huge. A possible solution is to attribute to each user an own radio-frequency beamforming matrix, involving users to be separated in time rather than frequency. In other words, it may be difficult to have a fully digital beamformer that is sub-carrier dependent due to hardware complexity constraints, while it is simpler to have a unique beamformer for the whole available Radio Frequency (RF) bandwidth that can change at the time slot rate [34].

Contrary to the conventional MIMO, massive MIMO systems can use linear precoders, such as MRT, MMSE, and ZF in order to reduce the implementation complexity [20].

ZF is a well-known combining algorithm which can also be applied in digital beamforming and can nullify the multi-user interference in a multi-user MIMO system. In this case, the ZF matrix $\mathbf{F}^{ZF}$ is calculated directly from the propagation channel $\mathbf{H}$ as follows:

$$\mathbf{F}_{BB}^{\mathrm{ZF}} = \mathbf{H}^H (\mathbf{H}\mathbf{H}^H)^{-1} \tag{29}$$

MMSE technique is an alternative where the principle of this method is to minimize the MMSE between the actual transmitted data and the received signal. The matrix associated with this optimization is:

$$\mathbf{F}_{BB}^{\mathrm{MMSE}} = (\mathbf{H}^H\mathbf{H} + 2\sigma_n^2\mathbf{I})^{-1}\mathbf{H}^H, \tag{30}$$

where $\sigma_n^2$ is the noise variance. Contrarily to ZF, the MMSE receiver promotes a better combination of interference reduction and noise enhancement, since it is designed to minimize the total noise.

The maximum ratio technique was employed in the transmission in [79] and studied as a receiving technique in [80]. The signals from all the antenna elements are weighted with respect to their SNR, being the optimum weights matched to the wireless channel. The amplitude is changed, and the phase of the individual signals must be adjusted, thus requiring an individual RF chain and phasing circuit for each antenna element. MR provides an output SNR equal to the sum of the individual SNRs, which produces the best statistical reduction of fading of any known linear diversity technique [81]. In MR, precoding matrix will be calculated as:

$$\mathbf{F}_{BB}^{\mathrm{MRC}} = \frac{\mathbf{H}^H}{T}. \tag{31}$$

In Equal Gain Transmission (EGT), all the received signals are transmitter with equal weights. The possibility of providing an acceptable signal from a few unacceptable inputs is still retained, and the performance is marginally inferior to MRT and superior to antenna selection, where only the highest SNR signal is considered for transmitting purposes. The EGT precoding matrix is given by

$$\mathbf{F}_{BB}^{\mathrm{EGT}} = \frac{exp\{jarg(\mathbf{H}^H)\}}{T}. \tag{32}$$

The performance assessment of these techniques employed on the receiver side was extended to MIMO scenarios in [80].

On the other hand, the analog beamforming of Figure 11 is a simpler and cheaper variant as the baseband is connected to the multiple antennas through phase shifters. These will be connected to each antenna so that it is possible to adjust only the signal phase. Phase shifters reduce the hardware limitations, allowing for low complexity implementations. However, the performance of fully analog beamforming techniques is limited, and it is usually only used for single-stream transmission. These constraints make it very difficult to form multiple beams, tune the sidelobes with accuracy, or steer the nulls.

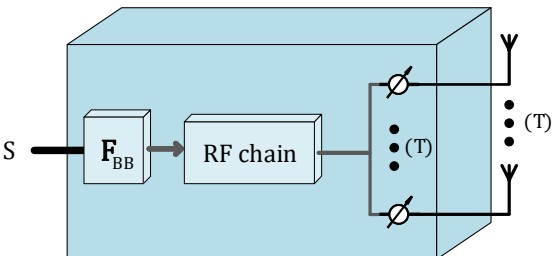

**Figure 11.** Analog beamforming structure.

The need for a suitable signal processing scheme for massive MIMO sparked the study and analysis between the digital and analog implementations trade-offs [82]. Hybrid beamforming as shown in Figure 12 have emerged as an approach which combines the best of both worlds by using a small number of RF chains and connect them to the antenna array through, for example, a stage of analog phase shifters [83–86]. This approach is motivated by the fact that the number of up-down conversion chains is only lower-limited by the number of data streams that are to be transmitted. This solution has attracted the attention from the academy and even the industry [87]. However, there is still the challenging task to design optimal hybrid beamforming schemes, which have a complex nature due to the non-convex constant modulo constrain imposed by the analog phase shifters [4,88–90]. In addition to the non-convex constrain, phase shifters are usually controlled digitally and have a discrete resolution. These conditions create a large number of possible combinations of beamforming weights and phases to be optimized, which carries a considerable computational complexity [91–93].

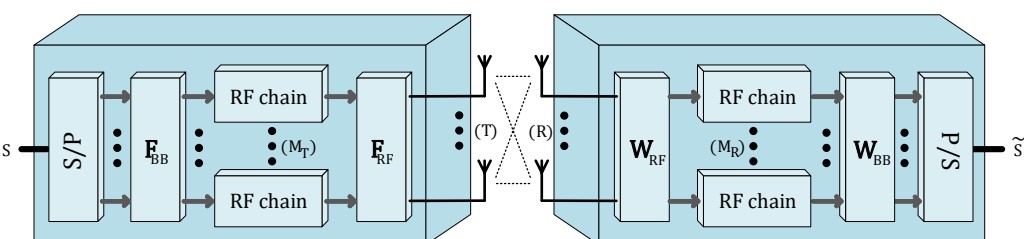

**Figure 12.** Hybrid MIMO transmission system structure.

Regarding the implementation of hybrid beamforming with analog and digital domains, two types of architectures have emerged: fully connected and sub-connected architectures shown in Figures 13 and 14, respectively. In the fully connected architecture, each RF chain is connected to all of the transmitter antennas through an analog device (switch, phase shifter, etc.). This architecture enables a bigger number of signal combinations and adjustments [94], but the optimization of the digital and analog precoder can have a high computation complexity.

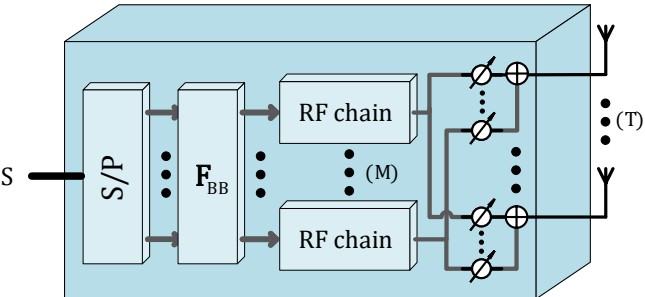

**Figure 13.** Hybrid beamformer with a fully-connected structure.

In the sub-connected architecture, each RF chain connects only a subset of antennas [95]. When compared with the fully connected counterpart, sub-connected architectures allow a smaller number of phase shifters. Thus, the power consumption is reduced, and the computational complexity is also lower. It is also important to mention that sub-connected architecture can be implemented in a dynamic or in a fixed way. In the dynamic sub-connected case, each RF chain can dynamically connect to a different set of antennas, and, in the fixed sub-connected one, each RF chain is always physically connected to the same set of antennas.

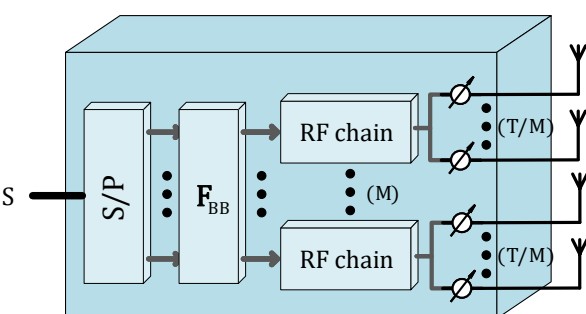

**Figure 14.** Hybrid beamformer with a sub-connected structure.

The analog phase sifters can be compacted in a system block, represented by a matrix $\mathbf{F}_{RF}$ whose entries are the complex values corresponding to the selected phase shifts. Despite the obvious advantages over the fully connected implementation, in a hybrid architecture, the CSI needs to be acquired through the hybrid 'lens' which presents a possible challenge.

*Multi-Layer Massive MIMO*

As the number of RF chains is increased, the efficiency of these power amplifiers becomes crucial to satisfy the energy efficiency requirements in massive MIMO systems. Thus, the use of NL power amplifiers in RF amplification stage allows for minimizing power consumption, being particularly important for high bit rate scenarios and critical at Gigabit data rates and above. However, to avoid signal distortion at NL amplifiers, the transmitted signal should have a low PAPR, which is difficult to achieve, especially in OFDM signals or SC signals using high order constellations and/or highly selective filtering. It is then essential to employ recoding and PAPR reduction techniques. However, as stated before, PAPR reduction techniques have limitations and are usually associated with an increased processing complexity and/or nonlinear distortion. Furthermore, using such power-efficient amplifier constraints can require the use of precoding techniques, whose complexity and power consumption might compromise the energy efficiency of the entire system. It was shown that SC modulation can, in theory, achieve near-optimal sum rate performance in massive MIMO systems operating with a low-transmit-power-to-receiver-noise-power ratios, distinct from the channel power delay profile and with an equalization-free receiver [96]. As already mentioned, SC modulation maintains an

almost constant envelope, yielding an optimal PAPR performance. Conventional MIMO systems can employ both NL precoding and linear precoding techniques without favoritism, although NL methods such as lattice-aided methods and dirty-paper-coding have better performance sacrificing the implementation low complexity [96].

A solution for the PAPR problem is given in [97], where a quasi-constant symbol decomposition is performed prior to the amplification stage. This system was further developed in [98], and the impact of different symbol decompositions was evaluated in [99].

Consider now the MIMO scenario depicted in Figure 8a characterized by a point-to-point communication link between a transmitter with $T = N_v \times N_m$ antennas and a receiver with $R \geq T$ receive antennas. The transmitter was designed with a layered structure as shown in Figure 15, aside from Layer 3, which is composed by a set of $N_v$ transmitters, each one composed by $N_m$ amplification branches in parallel (for QPSK and 16-QAM, we have $N_m = 2$ and $N_m = 4$, respectively). This means that the third layer has $T = N_v \times N_m$ antennas and transmits $N_v$ symbols' constellations simultaneously (more exactly, the $N_m$ components of each $N_v$ symbols from a QPSK or from a 16-QAM constellation, depending on the order of the original constellation).

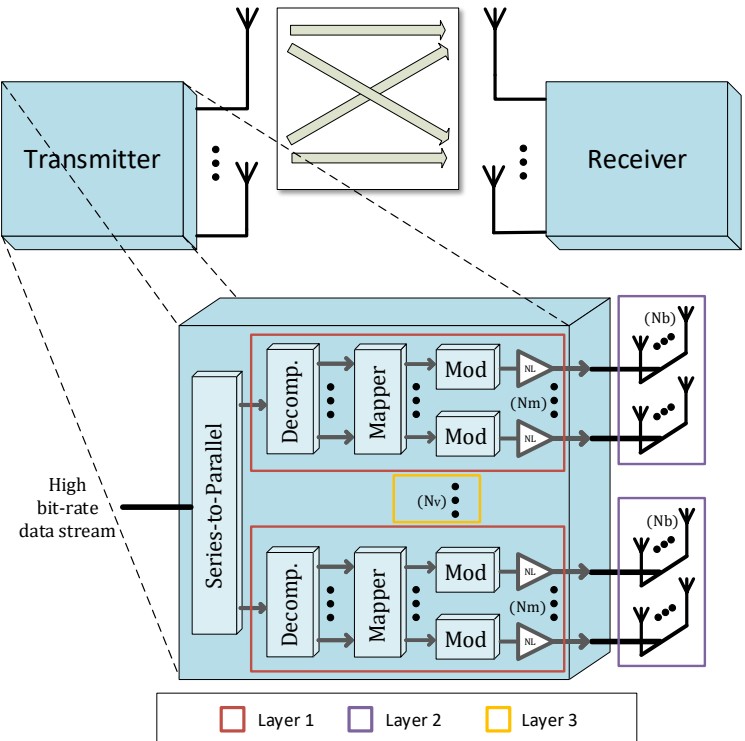

**Figure 15.** Layered MIMO transmitter structure.

Physical layer security can be also introduced by shaping of the transmitted constellation. Thus, the authorized user named "Bob" knows the transmitter configuration and the direction $\Theta$ (in which the constellation is optimized). On the other hand, the eavesdropper "Eve" will be unable to decode the information without the information about the constellation shaping. The influence of the channel is also excluded, since it is assumed that Eve may estimate the channel between the transmitter and Bob. Thus, any contribution from the channel to the security could be discarded (this situation can be generated in a scenario with a static channel or when the channel remains constant among several transmitted blocks, which is not the case of real wireless channel conditions). It is important to mention that we consider the unrealistic hypothesis that Eve could compute with a small error $\Delta\Theta$ the direction in which the constellation is optimized (this is only possible if Eve could estimate the transmitter's configuration, i.e., the spacing between antennas and the

arrangements of the components among the amplification branches, since the constellation shaping is a function of both factors).

A horizontal spacing of $\lambda/2$ between antennas of layer 1 and a vertical spacing of $\lambda$ between each set of $N_m$ antennas avoids the coupling effects among antennas. It is assumed that all antennas at transmitters are assigned to an authorized receiver with $R$ antennas. For comparison purposes, a transmitter with only spatial multiplexing with four antennas transmitting QPSK or 16-QAM constellations without any kind of shaping according to a specific direction $\Theta$ is also considered.

The channels between transmit and receive antennas are assumed to be time-dispersive and a SC-FDE block transmission technique is employed. To compensate the channel's frequency selectivity, an IB-DFE receiver can be adopted whose structure is depicted in Figure 16.

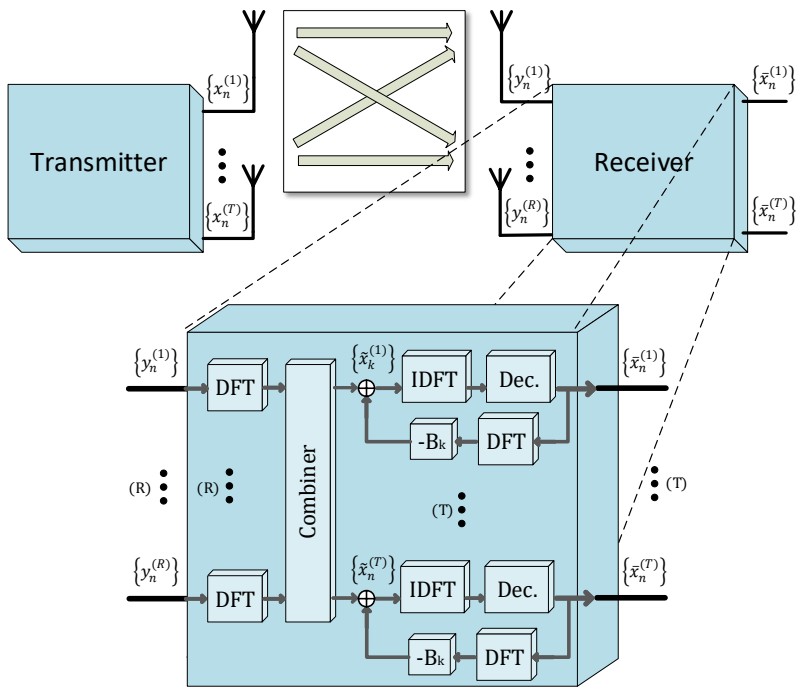

**Figure 16.** Overall massive MIMO system and details for SC-FDE receiver and equalizer.

At the transmitter, the $t$th antenna sends the block of $N$ data symbols $\{x_n^{(t)}; n = 0, 1, \ldots, N-1\}$, $\{y_n^{(r)}; k = 0, 1, \ldots, N-1\}$ being the received block at the $r$th receiver's antenna. In addition, a cyclic prefix with a length higher than the overall channel impulse response is appended to each transmitted block and removed at the receiver. Under these conditions, the corresponding frequency-domain received block $\{Y_k^{(r)}; k = 0, 1, \ldots, N-1\}$ is given by

$$\mathbf{Y}_k = \left[ Y_k^{(1)} \ldots Y_k^{(R)} \right]^T = \mathbf{H}_k \mathbf{X}_k + \mathbf{N}_k, \tag{33}$$

where $\mathbf{H}_k$ denotes the $R \times T$ channel matrix for the $k$th frequency, with $(r, t)$th element $H_k^{(r,t)}$, $\mathbf{X}_k = \left[ X_k^{(1)} \ldots X_k^{(T)} \right]^T$ and $\mathbf{N}_k$ denotes the channel noise.

For an iterative MMSE receiver, the data symbols for a given iteration can be obtained from the Inverse Discrete Fourier Transform of the block $\{\tilde{X}_k^{(t)}; k = 0, 1, \ldots, N-1\}$, where

$$\tilde{\mathbf{X}}_k = [\tilde{X}_k^1 \ldots \tilde{X}_k^{(R)}]^T = \mathbf{F}_k \mathbf{Y}_k - \mathbf{B}_k \overline{\mathbf{X}}_k, \tag{34}$$

where $\mathbf{I}$ is an appropriate identity matrix, and $\alpha = E[|N_k^{(r)}|^2]/E[|X_k^{(t)}|^2]$ is assumed to be identical for all antennas $t$ and $r$. Interference cancellation is done using $\overline{\mathbf{X}}_k = \left[ \overline{X}_0 \ldots \overline{X}_{N-1} \right]$,

with $\overline{X}_k$ denoting the frequency-domain average values conditioned to the FDE output for the previous iteration, which can be computed as described in [36]. Coefficients $F_k$, $B_k$, and the correlation coefficient $\rho$ can be computed as described in [100]. Since, in the first iteration, no information is available about the transmitted symbols and $\overline{\mathbf{X}}_k = \mathbf{0}$, this receiver can be regarded as a linear frequency-domain MMSE receiver. Next, iterations will employ the average values conditioned to the receiver output from previous iteration to remove the residual ISI.

The IB-DFE receiver has a computational complexity of $O(N_R{}^3)$ due to matrix inversions. To combat this issue, two other low-complexity iterative frequency-domain receivers, denoted as MRC and EGC, are also considered. As in [66], the ratio $R/T$ between the receiving and transmitting antennas is defined here to be at least equal to or higher than 4.

The MRC receiver is characterized there by

$$\tilde{\mathbf{X}}_k = \mathbf{\Psi}\mathbf{H}_k^H\mathbf{Y}_k - \mathbf{B}_k\overline{\mathbf{X}}_k, \tag{35}$$

and

$$\mathbf{B}_k = \mathbf{\Psi}\mathbf{H}_k^H\mathbf{H}_k - \mathbf{I}. \tag{36}$$

where $\mathbf{\Psi}$ denotes a diagonal matrix whose $(t,t)$th element is given by $\left(\sum_{k=0}^{N-1}\sum_{r=1}^{R}|H_k^{(r,t)}|^2\right)^{-1}$, takes advantage of the fact that

$$\mathbf{H}_k^H\mathbf{H}_k \approx R\mathbf{I}, \tag{37}$$

which is accurate when $R >> 1$ and the channels between different transmit and receive antennas have a small correlation. On the other hand, the EGC receiver characterized by

$$\mathbf{B}_k = \mathbf{\Psi}\mathbf{A}_k^H\mathbf{H}_k - \mathbf{I}. \tag{38}$$

where $\mathbf{\Psi}$ denotes a diagonal matrix whose $(t,t)$th element is given by

$$\left(\sum_{k=0}^{N-1}\sum_{r=1}^{R}|H_k^{(r,t)}|\right)^{-1}, \tag{39}$$

Taking advantage of the fact that, for massive MIMO systems with $R \gg 1$ and a small correlation between the channels associated with different transmit and receive antennas, the elements outside the main diagonal of

$$\mathbf{A}_k^H\mathbf{H}_k \tag{40}$$

are much lower than the values in the matrix diagonal.

It was already demonstrated that multilevel constellations can be decomposed in polar components, and the constellation symbols can be expressed as a function of the corresponding bits [44]. Let $\mathcal{S} = \{s_0, s_1, ..., s_{(N-1)}\}$, a constellation with $M$ points (i.e., $\#\mathcal{S} = M$), where $s_n \in \mathbb{C}$. To each constellation point $s_n$, we associate a set of $\mu = \log_2(M)$ bits in polar form $\mathcal{B} = \{b_n^0, b_n^1, ..., b_n^{(\mu-1)}\}$, with $b_n^{(i)} = \pm 1 = 2\beta_n^{(i)} - 1$, $\beta_n^{(i)} = 0$ or 1. The set of $\mu$ bits can be decomposed in $M = 2^\mu$ different subsets $\mathcal{B}_m$, $m = 0, 1, ..., M-1$.

Having $M$ constellation points in $\mathcal{S}$ and $M$ different subsets of $\mathcal{B}$, $\mathcal{B}_0, \mathcal{B}_1, ..., \mathcal{B}_{M-1}$, we can write

$$s_n = \sum_{m=0}^{M-1} g_m \prod_{b_n^{(i)} \in \mathcal{B}_m} b_n^{(i)}, n = 0, 1, ..., M-1 \tag{41}$$

which corresponds to a system of $M$ equations (one for each $s_n$ and $M$ unknown variables $g_m$). Without loss of generality, we can associate $m$ to its corresponding binary represen-

tation with $\mu$ bits, i.e., $m = (\gamma_{(\mu-1,m)}, \gamma_{(\mu-2,m)}, ..., \gamma_{(1,m)}, \gamma_{(0,m)})$ and define $\mathcal{B}_m$ as the set of bits where the bit $b_n^{(i)}$ is included if and only $\gamma_{(i,m)}$ is 1. Based on that, we may write

$$s_n = \sum_{m=0}^{M-1} g_m \prod_{i=0}^{\mu-1} (b_n^{(i)})^{\gamma_{(i,m)}}. \tag{42}$$

Figure 17 illustrated in detail the transmitter layered structure which combines the decomposition in polar components and a massive MIMO scheme with $N_v \times N_m \times N_b$ antenna elements, arranged in $N_v$ sets of $N_m \times N_b$ antennas.

Conventional beamforming schemes can be implemented by a layer 2 with $N_b$ antenna elements connected to each one of the $N_m$ amplification branches. A layer 3 with a set of $N_v \times N_m$ antennas is needed for spatial multiplexing, where the $N_m$ antennas of layer 1 are associated with the signal components of the constellation symbol and $N_v$ sets of $N_m$ antennas are assigned to simultaneously transmit $N_v$ multiple constellation symbols. The architecture in Figure 17 is an example of a transmitter based on layers 1, 2, and 3.

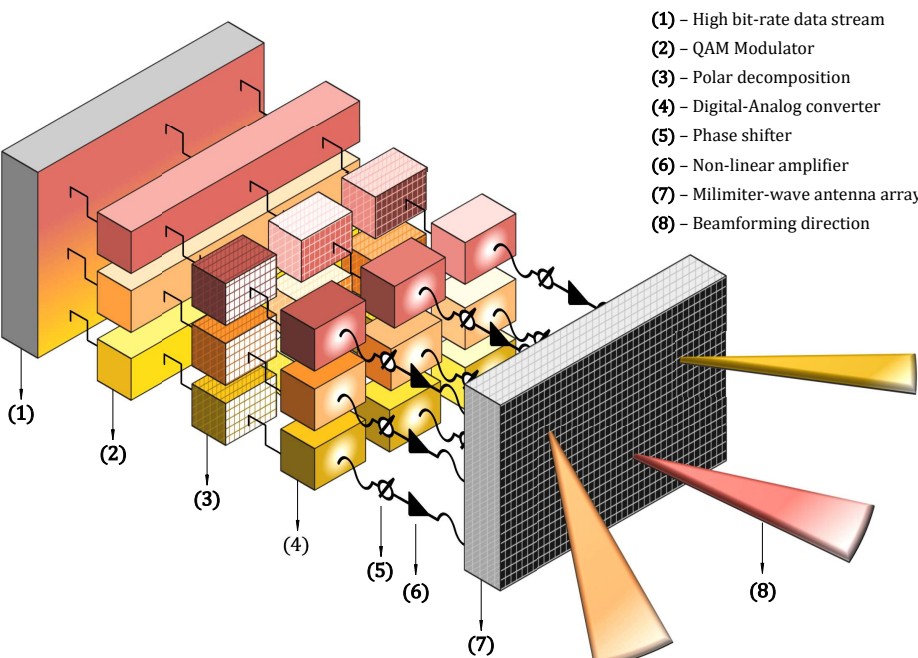

(1) – High bit-rate data stream
(2) – QAM Modulator
(3) – Polar decomposition
(4) – Digital-Analog converter
(5) – Phase shifter
(6) – Non-linear amplifier
(7) – Milimiter-wave antenna array
(8) – Beamforming direction

**Figure 17.** Transmitter layered structure.

Using this technique, power efficiency can be improved as the lower PAPR of the component signals allows for the adoption of NL amplifiers in layer 1 [101]. NL distortion effects were analyzed in [102] and channel estimation errors in [65]. Another essential aspect relies on the fact that each RF chain transmits uncorrelated signals, which means an omnidirectional radiation pattern for each set of $N_m$ antennas. However, each set of $N_m$ antennas implements directivity at the information level, since the constellation points of the transmitted signal maintain their positions at the desired direction $\Theta$ but are scrambled for other directions. It is important to mention that the phases of the NL amplifiers could be adjusted to assure information directivity using constellation shaping. Therefore, the authorized receiver can demodulate the received signal, while the undesired receivers cannot extract any useful information from the scrambled constellation. Hence, for a proper reception, the receiver needs to be in the right direction $\Theta_j$ and should know the transmitter parameters of Layer 1.

## 6. Promising Solutions

Regarding the massive MIMO scenario, multiple research works have explored the same path in the signal processing field by employing hybrid structures with digital and analog domains mostly due to the hardware cost and power consumption limitation. Although the digital processing was widely studied, the combination with an analog scheme is still rich in research challenges [103,104]. Different architectures for the analog domain were proposed with fully or sub-connected [105] schemes consisting of fixed phase shifters [86], dynamic phase shifters, and dynamic switches network [106], among others [107,108]. In addition, analog beamforming can be implemented in different ways as in [6] where the phase tuning is made using the FFT. Numerous solutions were proposed regarding distinct challenges or limitations:

- NL distortion effects [91,102];
- Energy efficiency [3,109];
- Employing low-resolution ADCs [110];
- Wide-band channels [111];
- Machine learning application [112];
- Channel estimation [66,92,113,114];
- Channel correlation [115];
- User clustering [116];
- System throughput maximization [117,118];
- Relay-assisted systems [89].

Other possible system solution can take advantage of the combination of the presented techniques. By considering a multi-layer massive MIMO architecture with a hybrid beamforming transmitter and a IB-DFE receiver structure, it is possible to optimize the hybrid precoder and the receiver parameters using optimization tools.

Accurate channel state information is fundamental in order to have good precoding and combining techniques. Channel Spatial orthogonality is compromised in several cases and, thus, we cannot rely on it in the signal processing. In addition, new approaches to the massive MIMO model, especially for the sparse scenario like mmWave transmission, are considering to re-build the transceiver design having as a basis the antenna arrays. For example, the high dimensional channel could be described by some chosen physical parameters, e.g., Angle of Arrival (AoA), Angle of Departure (AoD), multi-path delay, Doppler shift, etc. Therefore, transceiver techniques like synchronization, channel estimation, beamforming, precoding, multi-user access, etc., can be re-shaped with these physical parameters, as opposed to those designed directly with channel state information. Signal processing can be further exploited and provide enhanced performances in many aspects [68].

Low complexity quadrature spatial modulation (QSM) schemes for multi-user schemes were recently proposed in [54], where high spectral efficiency and low complexity were the main goals.

An approach regarding the optimization of a MIMO IB-DFE system was presented in [111], where the IB-DFE coefficients $F_k$ and $B_k$ were optimized considering the Minimum Mean Squared Error and using Lagrange multipliers.

Combining the multi-layer massive MIMO with a hybrid processing architecture arises as a promising solution that can be studied.

## 7. Conclusions

This article reviewed and analyzed massive MIMO systems and techniques from different perspectives as system capacity, hardware and computational complexity, and energy efficiency, and some remarks can be pointed out to summarize this work. Massive MIMO schemes can rely on the asymptotic linear independence when designing signal processing, considering the high number of employed antennas. Classic MIMO techniques are not able to meet the technical requirements, so hybrid architectures use a combination of analog and digital domains to exploit the spatial resolution provided by a large number

of antenna elements but keep the number of energy-hungry and expensive RF chains within reasonable limits and computational complexity controlled. It becomes clear that there is no single structure/algorithm that provides the best trade-off between complexity and performance in all the possible scenarios, but rather that there is a need to adapt them to application and channel characteristics in the system design. Regarding the optimization of the energy efficiency of RF power, it is achievable using solutions like multi-layer massive MIMO.

**Author Contributions:** Conceptualization, D.B., R.D., P.M.; methodology, D.B., R.D., P.M. software, D.B.; validation, D.B.; formal analysis, D.B.; investigation, D.B.; resources, P.M., M.B.; writing—original draft preparation, D.B.; writing—review and editing, R.D., P.M., M.B.; supervision, P.M., R.D., M.B.; project administration, P.M., M.B.; funding acquisition, P.M., M.B. All authors have read and agreed to the published version of the manuscript.

**Funding:** This work is supported by FCT/MCTES through national funds and when applicable co-funded EU funds under the projects MASSIVE5G (PTDC/EEI-TEL/30588/2017), PES3N (POCI-01-0145-FEDER-030629), Instituto de Telecomunicações (UIDB/EEA/50008/2020) and the grant (SFRH/BD/131093/2017) from Fundação para a Ciência e Tecnologia.

**Conflicts of Interest:** The authors declare no conflict of interest.

## Abbreviations

The following abbreviations are used in this manuscript:

| | |
|---|---|
| ADC | Analog-to-Digital Converter |
| AOA | Angle Of Arrival |
| AOD | Angle Of Departure |
| BPSK | Binary Phase Shift Keying |
| BS | Base Station |
| CDMA | Code Division Multiple Access |
| CP | Cyclic Prefix |
| DAC | Digital-to-Analog Converter |
| DFE | Decision Feedback Equalizer |
| DL | Downlink |
| EGC | Equal Gain Combiner |
| EGT | Equal Gain Transmitter |
| FDE | Frequency Domain Equalization |
| FFT | Fast Fourrier Transform |
| IB-DFE | Iterative Block Decision Feedback Equalizer |
| ICI | Inter Carrier Interference |
| IFFT | Inverse Fast Fourier Transform |
| IOT | Internet of Things |
| ISI | Inter Symbol Interference |
| LoS | Line-of-Sight |
| M2M | Machine to Machine |
| MC | Multi-Carrier |
| MIMO | Multiple-Input Multiple-Output |
| MISO | Multiple-Input Single-Output |
| MMSE | Minimum Mean Squared Error |
| mmWave | Millimiter Wave |
| MRC | Maximum Ratio Combiner |
| MRT | Maximum Ratio Transmitter |
| MU-MIMO | Multi-User Multiple-Input Multiple-Output |
| NL | NonLinear |
| NLoS | Non-Line-of-Sight |
| OFDM | Orthogonal Frequency Division Multiplexing |
| PAPR | Power to Average Power Ratio |
| PSK | Phase Shift Keying |

| | |
|---|---|
| QAM | Quadrature Amplitude Modulation |
| QoS | Quality of Service |
| QPSK | Quaternary Phase Shift Keying |
| RF | Radio Frequency |
| SC | Single Carrier |
| SDMA | Space Division Multiple Access |
| SIMO | Single-Input Multiple-Output |
| SISO | Single-Input Single-Output |
| SNR | Signal to Noise Ratio |
| STBC | Space Time Block Code |
| SVD | Singular Value Decomposition |
| TDD | Time-Division Duplexing |
| UE | User Equipment |
| UL | Uplink |
| ZF | Zero Forcing |

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
