# Peer review of "Massive MIMO Techniques for 5G and Beyond—Opportunities and Challenges"

_electronics, doi:10.3390/electronics10141667_

Round 1

Reviewer 1 Report

This paper addressing the Massive MIMO Techniques for 5G and Beyond - Opportunities and Challenges. The paper is good but it not addressing or pointed any thing about the spectrum extension by using the mmWave for 5G, I think authors should addressing or pointed on this point please consider such references for this point:

  • Saifullah Adnan, Yuli Fu , Naveed Ur Rehman Junejo, Zhen Chen, Hamada Esmaiel “Sparse Detection with Orthogonal Matching Pursuit in Multiuser Uplink Quadrature Spatial Modulation MIMO System” IET Communications, vol. 13, no 20, pp. 3472 – 3478, 2019.
  • Ahmed Abdelreheem, Ehab Mahmoud Mohamed, Hamada Esmaiel “Adaptive Location-Based Millimeter Wave Beamforming Using Compressive Sensing Based Channel Estimation” IET Communication, vol. 13, no. 9, pp. 1287-1296, 2019.
  • Ahmed S. Mubark, Hamada Esmaiel, Ehab Mahmoud Mohamed “LTE/Wi-Fi/mmWave RAN-Level Interworking Using 2C/U Plane Splitting for Future 5G Networks” IEEE Access, vol. 6, no. 1, pp. 53473-53488, 2018.
  • Ahmed Abdelreheem, Ehab Mahmoud Mohamed, Hamada Esmaiel “Location-Based Millimeter Wave Multi-level Beamforming using Compressive Sensing” IEEE Communications Letters, vol. 22, no. 1, pp. 185-188, Jan. 2018.
  • Ahmed S. Mubarak, Osama A. Omer, Hamada Esmaiel, and Usama S. Mohamed. "Geometry Aware Scheme for Initial Access and Control of MmWave Communications in Dynamic Environments" In Proc. of International Conference on Advanced Intelligent Systems and Informatics, Springer, Cham, pp. 760-769, 2019.
  • Ahmed Abdelreheem, Osama A. Omer, Hamada Esmaiel and Usama S. Mohamed “Location-Based Interference Cancellation in Device-to-Device Communications in Millimeter Wave Beamforming” In Proc. of 36th National Radio Science Conference (NRSC), IEEE, Egypt, 2019.
  • Ahmed S. Mubarak, Osama A. Omer, Hamada Esmaiel and Usama S. Mohamed “Backhaul Overhead Traffic Reduction in Dense MmWave Heterogeneous Networks Towards 5G Cellular Systems” In Proc. of 36th National Radio Science Conference (NRSC), IEEE, Egypt, 2019.
  • Ahmed M. Nor, Hamada Esmaiel, and Osama A. Omer. "Performance Evaluation of Proportional Fairness Scheduling in MmWave Network" In Proc. of International Conference on Computer and Information Sciences, IEEE, KSA, 2019.
  • Abdelreheem, Ahmed, Osama A. Omer, Hamada Esmaiel, and Usama S. Mohamed "Deep Learning-Based Relay Selection In D2D Millimeter Wave Communications" In Proc. of International Conference on Computer and Information Sciences, IEEE, KSA, 2019.
  • Ahmed S. Mubark, Ehab Mahmoud Mohamed, Hamada Esmaiel “Efficient mmWave Link Establishment and Maintaining using Wi-Fi/mmWave Interworking” in Proc. of IEEE, ICCECE’18, United Kingdom.
  •  

Reviewer 2 Report

The article is a summary of different MIMO techniques applied in mobile systems. I didn't find any novelty, but a simple summary which descibes the basic mathematical approaches of these methods. The article therefore can be considered as a introduction lecture note. The real problems are also not described in the article as antenna radiation characteristic imperfection effects, antenna mutual coupling effects, channel estimation methods.

Reviewer 3 Report

This paper reviewed and analyzed massive MIMO systems and techniques from different perspectives, such as system capacity, hardware and computational complexity and energy efficiency. Overall, the paper is well-written and easily understandable. However, the main concern is that there are similar surveys or tutorials in the literature, what is the main contribution of this paper? What are the new aspects that have not been reported before? There are not clear.

Besides, there are some typos in the manuscript, e.g., “Massive MIMO schemes can rely on the asymptotic linear independence when designing signal processing”, etc. Some figures are not well displayed, e.g., Fig. 1.

Round 2

Reviewer 1 Report

The author considered the reviewer's comments. 

Author Response

Thank you for your review !

Reviewer 2 Report

The authors corrected parts criticised by the reviewers, therefore can be published in Electronics. 

Author Response

Thank you for your review !

Reviewer 3 Report

Why take special attention to single-carrier schemes? For 5G and beyond, multi-carrier schemes are more favorable.

Author Response

Thank you for your remark. In fact 5G is expected to be centred on OFDM-based schemes. However, waveforms like DFT-S-OFDM, BWB-OFDM and GFDM include as special cases SC-FDE, with the associated advantages, especially for the uplink transmission. Moreover, massive MIMO is not restricted to 5G, and the advantages of SC-FDE-based mMIMO schemes apply to other scenarios. These issues were clarified in the revised version of our manuscript.